Familiar face + novel face = familiar face? Representational bias in the perception of morphed faces in chimpanzees

Matsuda Yoshi-Taka matsuda@brain.riken.jp 1
Myowa-Yamakoshi Masako 2
Hirata Satoshi hirata.satoshi.8z@kyoto-u.ac.jp 3
1 Center for Baby Science, Doshisha University, Kizugawa, Kyoto, Japan
2 Graduate School of Education, Kyoto University, Kyoto, Japan
3 Wildlife Research Center, Kyoto University, Kyoto, Japan
Vonk Jennifer
Electronic publication date: 2016 Aug 4
Publication date: 2016
Volume: 4
Electronic Location ID: e2304
Received 2015 Aug 10; Accepted 2016 Jul 7
Copyright: ©2016 Matsuda et al.
Copyright year: 2016
Copyright holder: Matsuda et al.
License: This is an open access article distributed under the terms of the Creative Commons Attribution License, which permits unrestricted use, distribution, reproduction and adaptation in any medium and for any purpose provided that it is properly attributed. For attribution, the original author(s), title, publication source (PeerJ) and either DOI or URL of the article must be cited.
License URL: https://creativecommons.org/licenses/by/4.0/

Keywords: Chimpanzee, Face perception, Novelty, Familiarity, Categorical perception, Preferential looking

Funding: Grants-in-Aid for Scientific Research KAKENHI 24000001 26245069 16H06283 24119005 24300103 This work was supported by funding from Grants-in-Aid for Scientific Research (KAKENHI 24000001, 26245069 and 16H06283 to SH; 24119005 and 24300103 to MM-Y). The funders had no role in study design, data collection and analysis, decision to publish, or preparation of the manuscript.

==============================
Highly social animals possess a well-developed ability to distinguish the faces of familiar from novel conspecifics to induce distinct behaviors for maintaining society. However, the behaviors of animals when they encounter ambiguous faces of familiar yet novel conspecifics, e.g., strangers with faces resembling known individuals, have not been well characterised. Using a morphing technique and preferential-looking paradigm, we address this question via the chimpanzee’s facial–recognition abilities. We presented eight subjects with three types of stimuli: (1) familiar faces, (2) novel faces and (3) intermediate morphed faces that were 50% familiar and 50% novel faces of conspecifics. We found that chimpanzees spent more time looking at novel faces and scanned novel faces more extensively than familiar or intermediate faces. Interestingly, chimpanzees looked at intermediate faces in a manner similar to familiar faces with regards to the fixation duration, fixation count, and saccade length for facial scanning, even though the participant was encountering the intermediate faces for the first time. We excluded the possibility that subjects merely detected and avoided traces of morphing in the intermediate faces. These findings suggest a bias for a feeling-of-familiarity that chimpanzees perceive familiarity with an intermediate face by detecting traces of a known individual, as 50% alternation is sufficient to perceive familiarity.

Introduction

Distinction between in- and out-group members based on faces has greatly influenced the evolution of society, leading to individualised relationships and cooperative networks (Hamilton, 1964). Notably, many social animals can distinguish between familiar and unfamiliar faces of conspecifics, such as humans (Bentin et al., 1996; Rossion, Schiltz & Crommelinck, 2003), chimpanzees (Boysen & Berntson, 1989; Fukushima et al., 2013); (Parr et al., 2000), orangutans (Hanazuka et al., 2013; Vonk & Hamilton, 2014), gorilla (Vonk & Hamilton, 2014), rhesus monkeys (Gothard, Erickson & Amaral, 2004; Murai et al., 2011), capuchin monkeys (Pokorny & De Waal, 2009), dogs (Racca et al., 2010), sheep (Kendrick et al., 2001), cattle (Coulon et al., 2009), pigeons (Wilkinson, Specht & Huber, 2010) and even invertebrates (Chittka & Dyer, 2012).

However, few studies have investigated how animals behave when they encounter ambiguous faces of familiar yet novel conspecifics, e.g., strangers with faces resembling known individuals. Theoretically, three behavioural outcomes are possible: animals could behave towards the ambiguous faces (1) in the same manner as the familiar faces, (2) in the same manner as the novel faces or (3) in a manner different from both the novel and familiar faces. A previous study showed that human adults perceive familiarity towards composite faces of novel and familiar (self, sibling and friend) individuals (Platek & Kemp, 2009), which supports the first possible outcome among the above-mentioned possibilities. This study suggests that facial resemblance acts as a cue for genetic relatedness (Daly, 1988) and increases ratings of trustworthiness (DeBruine, 2002). A bias towards novelty in responses to composite faces (the second possible outcome) is supported by the attention model of categorisation (Kruschke, 2003) or hypodescent (Halberstadt, Sherman & Sherman, 2011): attention is strongly tuned towards the features that best distinguish between in- and out-group members while ignoring shared features, i.e., the assignment of ambiguous faces to out-group members. We previously showed that human infants have a lower preference for composite faces, which include aspects of familiar and novel individuals, than original faces (Matsuda et al., 2012) (the third possibility discussed above). This study suggests that a negative feeling (or avoidant response) occurs because the composite faces elicit the concept of ‘familiarity’ but fail to satisfy it (e.g., Mori, 1970; Steckenfinger & Ghazanfar, 2009). It remains unknown whether other primates exhibit familiarity-biased responses, novelty-biased responses or negative responses when they encounter ambiguous faces. Answering this question may aid our understanding of the evolutionary and developmental profiles of facial recognition in detecting the category boundary between in- and out-group members.

We investigated this question in our closest living relative, the chimpanzee, using the morphing technique and a preferential looking paradigm. As is the case with human species, chimpanzees are known to possess a well-developed ability to readily individualise faces (Dahl et al., 2013; Hirata et al., 2010; Kano & Tomonaga, 2009; Tomonaga, 2007) and distinguish between familiar versus novel conspecifics (Fukushima et al., 2013) even when shown only static two-dimensional images that are absent any other cues (e.g., olfaction). They also have the ability to identify individuals in terms of kinship detection via phenotypic matching (Parr & De Waal, 1999; Parr et al., 2010). Furthermore, chimpanzees show categorical perception in face recognition; that is, they detect a category boundary on a spectrum-ranging morphed continuum that comprises two different faces, and their sensitivity for categorisation is affected by exposure to conspecific/non-conspecific faces during development (Martin-Malivel & Okada, 2007). The morphing technique has the advantage of creating composite portraits between two faces with physical accuracy in blend ratios, which enables us to manipulate and quantify the degrees of similarity between different faces (DeBruine et al., 2008; Matsuda et al., 2012; Myowa-Yamakoshi et al., 2005; Parr et al., 2012).

In contrast to the discrimination paradigm, the preferential looking paradigm with free viewing has the advantage of investigating intrinsic gazing behaviours such as preference, attention or processing costs by minimising training effects and testing beyond the limits of a match-to-sample task, as we reported previously in adult and infant chimpanzees (Hirata et al., 2010). The aim of our study was to quantify the degree of similarity between faces that chimpanzees perceive as belonging to familiar conspecifics. We investigated the gazing behaviour of chimpanzees when presented with composite faces representing morphs between familiar and novel conspecifics. First, we presented ‘intermediate faces’ (50% familiar and 50% novel faces) to chimpanzees and examined whether they preferred these intermediate conspecific faces to the original faces. We identified the faces preferred by the chimpanzee when presented with three pairs of faces: familiar versus novel faces, familiar versus intermediate faces and novel versus intermediate faces.

Given that novel faces (i.e., potential threats) should elicit more of a response from the chimpanzees than familiar faces (Pascalis & Bachevalier, 1998), we measured the time (fixation duration) spent looking at a face, the frequency (fixation count) with which the chimpanzee looked at the face, and the saccade length for how extensively they scan the face; these indices were used to measure the attentional and processing demand/efficacy (i.e., informativeness) (Antes, 1974; Henderson, 2003; Henderson & Smith, 2009; Loftus & Mackworth, 1978; McCamy et al., 2014; Unema et al., 2005) during facial recognition and to differentiate gazing behaviours associated with viewing familiar and novel faces.

Methods

This research was conducted in accordance with the Guide for the Care and Use of Laboratory Animals of Hayashibara Biochemical Laboratories, Inc. and the Weatherall Report, The use of non-human primates in research. The research protocol was approved by the Animal Welfare and Animal Care Committee of the Hayashibara Great Ape Research Institute (GARI-051101).

Participants

Eight chimpanzees (Pan troglodytes, two males and six females, 4–17 years old) participated in this study. The names, ages, and sexes of the chimpanzees were shown in Table 1. The subjects were members of a captive group at the Great Ape Research Institute (Hayashibara Biomedical Laboratories, Inc.). Relative dominances and kinship structures were shown in Tables S1 and S2. Loi, Zamba, Tsubaki, and Mizuki came to the institute from other locations when they were 3, 3, 3, and 2 years old, respectively, and since arriving have spent their time together in the same group. Misaki was added to the group when she was 3 years old. Since then, she has spent her time with the above-mentioned individuals in the same enclosure, with few exceptions (described below). Natsuki was born to Tsubaki, and Iroha was born to Mizuki. These offspring grew up in the same group with their mothers and other group members. Hatsuka is a daughter of Misaki but was neglected between the ages of 40 days and 2.5 years. During this period, human caretakers nursed Hatsuka, but she spent 10 min to 3 h of nearly every day with all of the other chimpanzees in the same space. Misaki spent all of her time with the other chimpanzees (except her daughter) during this period. After the age of 2.5 years, Hatsuka spent all of her time with Misaki (her mother), and the two of them were separated from the other members in an adjacent enclosure. However, they maintained visual contact with the other members in the neighbouring enclosure through wire mesh or transparent panels. In short, the adult participants (Loi, Zamba, Tsubaki, Mizuki, and Misaki) had known each other since they were 2–3 years old, for approximately 10 years or more, and the adolescent (Natsuki) and infant (Hatsuka and Iroha) individuals were born into this group and had known each other and the other group members since they were born.

Table 1 Participants’ names, age and sex.

Reference number	Name	Age (years)	Sex	
Chimpanzee 1	Hatsuka	4	Female	
Chimpanzee 2	Iroha	4	Female	
Chimpanzee 3	Loi	17	Male	
Chimpanzee 4	Misaki	12	Female	
Chimpanzee 5	Mizuki	15	Female	
Chimpanzee 6	Natsuki	7	Female	
Chimpanzee 7	Tsubaki	16	Female	
Chimpanzee 8	Zamba	17	Male	

Three individuals, whose faces were presented to the subjects as familiar-face stimuli, also participated in the study (Loi, Misaki and Zamba). Although these three individuals saw their own faces on a screen in the experiment, our previous study using the electroencephalography (EEG) recording showed that no significant difference was observed between self- and familiar-face recognition (Fukushima et al., 2013). This finding may be caused by past experiences with informal tests of mirror self-recognition and recognition of self-images on TV monitors (S Hirata, K Fuwa & M Myowa-Yamakoshi, 2015, unpublished data). Thus, we presented self-face stimuli as ‘familiar’ faces to the three individuals. This is consistent with a previous study that showed animals failed to treat self images any differently than other familiar faces, which suggests that the subjects found self images to be familiar given the glass and other reflective surfaces present in their outdoor habitats (Vonk & Hamilton, 2014).

The relative dominances and kinship structures between each subject and familiar-face stimuli were tabulated (Tables S1–S2).

Experimental apparatus

The subjects sat in an experimental room (3 m × 2 m × 2.5 m, L × W × H) and viewed the images on a 17-inch LCD screen (1,024 × 768 pixels) at a distance of 60 cm. The eye movements of the chimpanzees were recorded using a table-mounted eye tracker (Tobii T60, Stockholm, Sweden) (Hirata et al., 2010).

Stimuli description

Prior to the experiments, coloured and frontal-orientation photographs of familiar (in-group) chimpanzees and novel chimpanzees (from an out-group housed at the Kumamoto Sanctuary, Wildlife Research Centre, Kyoto University) were taken. Each photograph was rotated in-plane to horizontally align the interpupil distance and was then resized to 512 × 680 pixels (at a distance of 60 cm with approximately 13.3 × 17.4 degrees of visual angle). Using computer morphing software (Sqirlz Morph 2.1: Xiberpix, Solihull, UK, www.xiberpix.com), approximately 350 points were positioned on each chimpanzee face (approximately 100 points for facial contour and 250 for facial parts) to delineate specific facial landmarks (e.g., distinctive wrinkles under the eyes, across the muzzle, and in the ear region) (Debruine et al., 2008; Parr et al., 2012). To create intermediate faces, a familiar face and a novel face were morphed together using the software (Sqirlz Morph 2.1) to produce a new face incorporating 50% of the familiar face and 50% of the novel face by calculating the mean shape and colour of the constituents (Fig. 1) (Matsuda et al., 2012). A previous study showed that chimpanzees could detect facial similarities between mothers and sons but not between mothers and daughters (Parr & De Waal, 1999), which suggests that chimpanzees are better at detecting facial similarities among males; consequently, we morphed two photographs of male chimpanzees to create an intermediate face. Intermediate faces were created from photographs of different chimpanzees than those used as the familiar face and novel face stimuli; this step was taken to prevent an adaptation effect that could occur when the chimpanzees were repeatedly presented with images of the same faces (even though the morphs bore only partial resemblance to the original faces).

Figure 1 Visual preferences of chimpanzees for different types of faces.

(A) An example of three different types of stimuli: familiar face (i), intermediate face (ii) and novel face (iii). (B) The mean-proportional fixation duration for each of the face types. (C) The mean-proportional fixation count for each of the face types. (D) The mean-proportional saccade length for each of the face types. Boxplots describe the responses to each type of facial stimuli. *p < 0.05, **p <0.01 (Friedman test).

We used movie clips of dynamic facial expressions as visual stimuli because primates and humans are more responsive to moving faces than to static faces (Shepherd et al., 2010). We created moving stimuli of mouth-opening chimpanzee faces that were familiar, novel or intermediate; these stimuli are known as dynamic facial expressions or dynamic faces (Kilts et al., 2003; LaBar et al., 2003). Moving stimuli were created in the following manner: first, two coloured photographs (mouth-closing and mouth-opening faces) of each individual chimpanzee were taken prior to the main experiment. Next, for each participant, ten intermediate images with expressions situated between the close-mouthed and open-mouthed expressions were created in 9% steps using the computer-morphing techniques. Then, to create a moving clip, the 12 images (one mouth-closing image, 10 intermediate images and the final mouth-opening image) were presented in succession. Each image was presented for 40 ms, and the final image was presented for an additional 760 ms; thus, each animation clip lasted 1,200 ms. Each clip was shown five times (i.e., totalling a six-second duration) in both the main and control experiments (see video clips of examples in Movies S1–S3; familiar-face.avi, intermediate-face.avi and novel-face.avi). A human expert viewed the images to confirm that the presentation speed sufficiently reflected the natural changes in the dynamic facial expressions of chimpanzees.

Experimental procedure and data analysis

Each chimpanzee’s gaze was calibrated on the eye tracker prior to the experiment. Two-point automated calibration was conducted by presenting a movie clip on each reference point. A relatively small number of reference points was adopted for the chimpanzees because they tended to view these reference points only briefly and no training procedure was adopted for them. However, we checked the accuracy after the initial calibration and repeated the calibration if necessary. Our validation session confirmed the comparable accuracy between chimpanzees and humans (see Hirata et al., 2010; Kano et al., 2011; the calibration errors were 0.2–1 degree in both chimpanzees and humans). Then, the chimpanzees were presented with the following four pairs of stimuli: (1) familiar versus novel faces, (2) familiar versus intermediate faces, (3) novel versus intermediate faces and (4) novel versus morphed faces of two novel chimpanzees. In each of the four trials, a pair of faces was presented side-by-side on an eye tracker screen for six seconds. Each novel chimpanzee face was presented only once to prevent an adaptation effect. Each trial was preceded by a stimulus intended to attract the participant’s visual attention to the centre of the screen. The order of the four test trials and the side on which a given face appeared was random and counterbalanced across participants.

In the data analysis, entire faces of presented stimuli were chosen as areas of interest rather than specific facial parts because we used movie stimuli of dynamic facial expressions as mentioned above. We measured the total looking durations, counts and saccade lengths for each face. The saccade length was measured as a sum of the looking path length during the face scanning for each face. We averaged the same types of facial stimuli for each participant to minimise the variations in looking behaviours that occasionally appear during free viewing. The data were normalised to calculate proportions between three types of facial stimuli (familiar, intermediate and novel faces for Fig. 1) and between two types of facial stimuli (novel and morphed faces of two novel faces for Fig. 2) (Matsuda et al., 2012). We omitted data if the chimpanzee (participant) looked at only one side of a pair of stimulus faces (strong side bias; see chimpanzee #4 eye tracking data in the supplementary information for Fig. 2).

Figure 2 Visual preferences of chimpanzees for different face types.

(A) The mean proportional fixation duration for each of the face types: a 100% novel face and a 50%–50% morphed face with different novel faces. (B) The mean proportional fixation count for each of the face types. (C) The mean proportional saccade length for each of the face types. Boxplots describe the responses to each type of face stimuli. n.s.: no significant difference (Wilcoxon signed-rank test).

Results

Figure 1A depicts an example of the three different types of stimuli: a familiar face, an intermediate face and a novel face. The fixation duration, fixation count and saccade length for each image are shown in Figs. 1B–1D, respectively. The non-parametric Friedman test for all participants (N = 8) revealed significant overall effects for the fixation duration (χ2(2) = 6.25, N = 8, p = 0.04; Fig. 1B), for the fixation count (χ2(2) = 7.47, N = 8, p = 0.02; Fig. 1C) and for the saccade length (χ2(2) = 9.25, N = 8, p = 0.01; Fig. 1D). The non-parametric Wilcoxon signed-rank test (two-tailed) showed that the fixation count and saccade length were significantly different between the familiar and novel faces (Z = − 2.24, N = 8, p = 0.02 for the fixation count and Z = − 2.52, N = 8, p = 0.01 for the saccade length), whereas the fixation duration was similar (Z = − 1.68, N = 8, p = 0.11); all indices were significantly different for the intermediate and novel faces (Z = − 2.52, N = 8, p < 0.01 for the fixation duration; Z = − 2.24, N = 8, p < 0.03 for the fixation count; Z = − 2.38, N = 8, p < 0.02 for the saccade length). No significant differences in these indices were found for the familiar and intermediate faces (Z = 0.70, N = 8, p = 0.55 for the fixation duration; Z = − 0.35, N = 8, p = 0.78 for the fixation count; Z = 0.14, N = 8, p = 0.95 for the saccade length). These results suggest that novel faces elicit a greater response than familiar or intermediate faces.

Although intermediate faces were less preferred than novel faces, it is possible that the chimpanzees detected traces of morphing (e.g., a blur in wrinkles) and subsequently avoided the unnatural-looking intermediate faces. In this case, the chimpanzees should show a lower preference for the morphed faces per se, irrespective of constituent individuals or whether a familiar or novel face was shown. To exclude this possibility, we presented the subjects with an image of a novel chimpanzee and an image of a morphed face between two different novel chimpanzees. The duration spent looking at the two faces and the frequency with which the faces were observed did not differ significantly, nor did the saccade length for facial scanning (Wilcoxon signed-rank test (two-tailed), Z = 1.52, N = 7, p = 0.16 for the fixation duration, Fig. 2A; Z = 1.18, N = 7, p = 0.30 for the fixation count, Fig. 2B; Z = 0.68, N = 7, p = 0.58 for the saccade length, Fig. 2C). This result indicates that the participants neither detected nor avoided the morphed faces; instead, they showed a lower preference for the intermediate faces than the novel faces because of constituent familiar faces.

We found systematic differences for participants in the perception of facial parts and body (the eyes, mouth and body of stimuli were defined as areas of interest); the non-parametric Friedman test for all participants revealed significant overall effects for the fixation duration (χ2(2) = 14.25, N = 8, p = 0.0008). The non-parametric Wilcoxon signed-rank test (two-tailed) showed that the fixation duration was significantly different between the eyes and mouth (Z = − 2.52, N = 8, p = 0.008) and between the mouth and body (Z = 2.52, N = 8, p = 0.008) but not between the eyes and body (Z = 1.40, N = 8, p = 0.20). Larger attention to the mouth part of facial stimuli may result from the presentation of movie stimuli of mouth-opening faces.

Discussion

In contrast with our previous human study (Matsuda et al., 2012), chimpanzees did not exhibit negative/ avoidant responses to intermediate faces (i.e., 50–50% composite faces between familiar and novel conspecifics). This difference occurred largely because the chimpanzees paid less attention to the familiar faces and preferred the novel faces. This novelty preference for familiar faces appeared as early as approximately 200 ms after stimulus presentation (Data S1), which is consistent with our previous EEG study that showed that different neural responses to familiar and novel faces have a latency of approximately 200 ms (Fukushima et al., 2013).

Furthermore, the chimpanzees appeared to avoid looking at the familiar and intermediate faces to a similar extent. One possible reason for this outcome was that morphing made a face more similar to the prototype and, therefore, made the face more attractive (Langlois & Roggman, 1990); however, this was not the case. Rather, the results indicate that the chimpanzees detected the resemblance between the intermediate face and the familiar face and thus responded to these faces in a similar manner. In other words, the category boundary between familiar (i.e., in-group) and novel (i.e., out-group) faces is more closely situated towards novel faces on a spectrum ranging from the familiar to the novel. This claim is consistent with a previous study of morphed-face recognition in humans that showed a bias in favour of a feeling of familiarity when one encounters intermediate faces (50–50%) of self-, sibling- and friend-stranger morphing (Platek & Kemp, 2009).

The ability to detect facial resemblance may play a key role in kinship recognition. Previous studies illustrated that chimpanzees are capable of visually recognising kin from pictures of novel individuals (Parr & De Waal, 1999; Parr et al., 2010). These previous studies, however, used a matching-to-sample paradigm, and the chimpanzees went through basic training of the match-to-sample task for familiar related individuals; thus, they may have simply learned to detect individuals with facial resemblances (Parr & De Waal, 1999; Parr et al., 2010). Using a free-viewing paradigm, we demonstrated that chimpanzees voluntarily detect individuals with facial resemblances.

We previously showed that human infants have a lower preference for intermediate faces (i.e., morphs between familiar (mother) and novel (stranger) faces) than for original faces (Matsuda et al., 2012), which is different from the results of the chimpanzees in the present study. This variation may originate from differences in the species, the age of the chimpanzees and/or the target faces. Thus, it is difficult to directly compare our results from human infants and adult chimpanzees without considering species similarities and differences in their cognitive development and adaptive significance. For example, a previous study showed that infant chimpanzees preferentially looked at their mother’s face, the most familiar face to the infants, over a computer-created, average face (i.e., novel face) when they were between 4 and 8 weeks of age; however, such a preference for the mother’s face suddenly disappeared after 8 weeks of age (Myowa-Yamakoshi et al., 2005). The developmental change in the mother/stranger preference has also been observed in human infants (Gredeback, Fikke & Melinder, 2010). This may be because both chimpanzee and human infants need to recognise their caregiver during the initial phase of their development (it is vital for their survival); however, infants become socialised at a certain stage by engaging in interactions with individuals other than their mothers (Matsuzawa, 2006). In short, it is natural to assume that preferences for certain faces are affected by various factors, including development and the environment (Dahl et al., 2013; Martin-Malivel & Okada, 2007).

The present study revealed a facial preference pattern in chimpanzees using familiar, novel, and morphed intermediate faces as stimuli in a free viewing paradigm. Our study suggests a representation bias in favour of a feeling of familiarity when one encounters intermediate faces between in- and out-group conspecifics.

Supplemental Information

Supplemental Information 1 Supplementary materials, eye-tracking data and cumulative histogram of fixation duration

Click here for additional data file.

Data S1 Raw data of eye tacking (Excel file)

Raw data of all participants (eight chimpanzees) and analyzed data for figures 1 and 2.

Click here for additional data file.

Movie S1 Familiar-face movie clip (example)

Click here for additional data file.

Movie S2 Novel-face movie clip (example)

Click here for additional data file.

Movie S3 Intermediate-face movie clip (example)

Click here for additional data file.

Additional Information and Declarations

Competing Interests

Author Contributions

Animal Ethics

Data Availability

The authors declare there are no competing interests.

Yoshi-Taka Matsuda conceived and designed the experiments, analyzed the data, contributed reagents/materials/analysis tools, wrote the paper, prepared figures and/or tables.

Masako Myowa-Yamakoshi conceived and designed the experiments, contributed reagents/materials/analysis tools, reviewed drafts of the paper.

Satoshi Hirata performed the experiments, contributed reagents/materials/analysis tools, reviewed drafts of the paper.

The following information was supplied relating to ethical approvals (i.e., approving body and any reference numbers):

This research was conducted in accordance with the Guide for the Care and Use of Laboratory Animals of Hayashibara Biochemical Laboratories, Inc. and the Weatherall report, The use of non-human primates in research. The research protocol was approved by the Animal Welfare and Animal Care Committee of the Hayashibara Great Ape Research Institute (GARI-051101).

The following information was supplied regarding data availability:

The raw data has been supplied as a Supplementary File.

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
