# Peer review of "Familiar face + novel face = familiar face? Representational bias in the perception of morphed faces in chimpanzees"

_PeerJ, doi:10.7717/peerj.2304_

## Round 0.1 · original submission · Minor Revisions

This is a very interesting, innovative, and well-designed study. I agree with Reviewer 2 that the methods are exceptionally clearly described and rigorous. I would suggest only minor improvements in terms of the background as per the suggestions of Reviewer 1. Some of the literature on recognition of familiarity in apes in monkeys is not cited. We recently published a paper on the recognition of familiar versus unfamiliar individuals in photographs with orangutans and a gorilla (Vonk and Hamilton, 2014) in which we review other relevant recent work. We also presented self stimuli in our tests and found no significant differences, which I think is interesting and may pose problems for the idea that the apes are classifying familiar stimuli as familiar in-group members. I think this issue should be discussed a bit further. Anna Wilkinson has also shown that pigeons make such distinction between familiar and novel individuals from photographs. I would like if your introduction laid more of the groundwork to show that this ability exists even when showing only static two dimensional images that are absent any other cues (olfaction etc.) based on the existing literature.
I would also be curious as to whether age or sex had any impact on the results.

Reviewer 1 ·

Basic reporting

The presented study shows a bias in the representation of in- and out-group faces in chimpanzees. Chimpanzees treat a 50% novel and 50% familiar face as familiar, indicated by greater looking preferences toward the novel face as opposed to the ‘morphed’ face.

(1) Title: “Familiar face + novel face = familiar face?” Confusing… Maybe change this to something more scientific, like: “Representational bias in the perception of…”
(2) Introduction: Uncanny Valley: page 5 line 12: This statement is wrong: Steckenfinger and Ghazanfar, 2009, PNAS showed the uncanny valley in monkeys. If uncanny valley is discussed in the current manuscript, then included Steckenfinger and Ghazanfar. However, it is not clear why in the presented work an uncanny valley may appear. Results don’t show an uncanny valley, and stimuli were not designed to elicit an uncanny valley, from what I understood. Instead of discussing the uncanny valley, I would add more content to the representational structure underlying familiar and novel faces.
(3) Representational bias: As suggested in (1), the main conclusion is that there is a representational bias in the perception of faces toward novel faces (longer looking times). 50% morphs are perceived as closer to familiar faces than novel faces. Representational biases as this can be described with simple computational considerations and appear at various levels of face processing (Race, Species, Gender, etc); see Furl, Phillips and OToole, 2002 or Dahl, Chen and Rasch, 2014. On a neural level Sigala, Logothetis and Rainer, 2011 describe an important concept, consistent with the current findings. The authors mention the boundary lines between in and out group members (page 20, line3ff). I would expand this discussion and make it the main conclusion.
(4) Generally, the manuscript is focusing on chimpanzee literature. Primate face perception, however, has been conducted primarily in monkeys for four decades, like the examples above.

Experimental design

(1) Mentioned are the origins of individual chimpanzees. How was the looking behavior influenced by their background (wild vs captivity)?
(2) Did the authors find systematic differences in the perception of facial parts by looking at areas of interests?
(3) Stimuli were movie sequences of mouth opening, which to some extent might be considered as emotional expression or communication signal. Did the authors look at temporal changes of dependent variables?
(4) Page 22, lines 4-5: In context of development and environment it would be interesting to discuss changes (if there are) of the looking strategies across different age classes. There is evidence in the chimpanzee literature on face perception that the age of the chimpanzee, or the experience with face classes, respectively, influences the perception of faces (Martin-Malivel and Okada, 2007; Dahl et al, 2013).

Validity of the findings

(1) Page 22, lines 6ff: I am not sure if I understand the importance these findings have on the evolutionary understanding of facial recognition. Is it that chimpanzees show representational biases of face classes? Although in context with recent chimpanzee studies mentioned above and lots of monkey studies that would not be new and important. Or is it the fact that chimpanzees can tell apart in- from out-group by the face? This statement needs explanation.

Additional comments

This finding per se is interesting and matches well with what is known in the literature in primate face perception. However, the manuscript needs some more detailed literature review in introduction and discussion as well as some more statistical comparisons. I generally like this study and support it for publication, once these minor issues have been taken care of.

·

Basic reporting

No comments

Experimental design

No Comments

Validity of the findings

No Comments

Additional comments

Overall, the manuscript is carefully and clearly written, making sure to connect findings to previous work and not draw over-arching conclusions in terms of comparisons with results in human infants. Too many researchers in our field (broadly speaking, comparative psychology) draw comparisons between their work and results with other species without considering the methodological, ontological, evolutionary, and cognitive differences that could be present. These authors took these possibilities into consideration while discussing their conclusions, indicating an understanding of the pitfalls of drawing comparisons when too many factors are left uncontrolled.

---

## Round 0.2 · Minor Revisions

Thank you for making the changes requested by the reviewers and for including more background literature in your Introduction. I believe the paper will be acceptable for publication with the following minor changes:

Vonk and Hamilton (2014) studied orangutans and one gorilla. Please cite appropriately on lines 8-10, p.3. We did not test rhesus macaques.

On line 11, support should be “supports”

Some of the paragraphs are still too long. Please break up the long paragraph that spans pg. 6-9. You could start one new paragraph with “In contrast to the discrimination paradigm..” You could start another with “Given that novel…”

Please create one Table (not in supplemental) to describe the names, age and sex of the chimpanzees and do not describe in detail in the text OR remove the tables and the reference to chimpanzee # and just list in the text with sex and age in years in parentheses.
On line 9, p.11 “nearly” is misspelled.

The Vonk & Hamilton study didn’t provide evidence that individuals fail to recognize self images as familiar (as cited on lines 11-13, p.12). They just failed to treat self images any differently than other familiar faces – a finding that appears consistent with your own findings.

Please ensure that your MS is consistent in American versus British spelling. For example, on p. 13, line 9 and p. 14 line 5, “coloured” is spelled with the British spelling but you use “behavior” more commonly. Check for all instances of colour and favour.
Please indicate that all of your tests were two-tailed, if so. If not, they should be two-tailed.

Please check line 15 on pg. 24 where there appears to be an unaccepted correction and an unneeded space in “environment”.

---

## Round 0.3 · accepted · Accept

Thank you for so promptly attending to the last remaining corrections. I am happy to accept your very interesting MS for publication now.